# A Dictionary Optimization Method for Reconstruction of ECG Signals after Compressed Sensing

**DOI:** 10.3390/s21165282

**Published:** 2021-08-05

**Authors:** Luca De Vito, Enrico Picariello, Francesco Picariello, Sergio Rapuano, Ioan Tudosa

**Affiliations:** Department of Engineering, University of Sannio, 82100 Benevento, Italy; epicariello@unisannio.it (E.P.); fpicariello@unisannio.it (F.P.); rapuano@unisannio.it (S.R.); ioan.tudosa@unisannio.it (I.T.)

**Keywords:** compressed sensing, dictionary learning, ECG, OMP

## Abstract

This paper presents a new approach for the optimization of a dictionary used in ECG signal compression and reconstruction systems, based on Compressed Sensing (CS). Alternatively to fully data driven methods, which learn the dictionary from the training data, the proposed approach uses an over complete wavelet dictionary, which is then reduced by means of a training phase. Moreover, the alignment of the frames according to the position of the R-peak is proposed, such that the dictionary optimization can exploit the different scaling features of the ECG waves. Therefore, at first, a training phase is performed in order to optimize the overcomplete dictionary matrix by reducing its number of columns. Then, the optimized matrix is used in combination with a dynamic sensing matrix to compress and reconstruct the ECG waveform. In this paper, the mathematical formulation of the patient-specific optimization is presented and three optimization algorithms have been evaluated. For each of them, an experimental tuning of the convergence parameter is carried out, in order to ensure that the algorithm can work in its most suitable conditions. The performance of each considered algorithm is evaluated by assessing the Percentage of Root-mean-squared Difference (PRD) and compared with the state of the art techniques. The obtained experimental results demonstrate that: (i) the utilization of an optimized dictionary matrix allows a better performance to be reached in the reconstruction quality of the ECG signals when compared with other methods, (ii) the regularization parameters of the optimization algorithms should be properly tuned to achieve the best reconstruction results, and (iii) the Multiple Orthogonal Matching Pursuit (M-OMP) algorithm is the better suited algorithm among those examined.

## 1. Introduction

In recent decades, technological advancements have led to the implementation and diffusion of Wearable Health Devices (WHDs). These devices can measure and acquire, in real time and without using bulky equipment, clinical information about a human user, such as: (i) ElectroCardioGram (ECG), (ii) blood pressure, (iii) respiration wave, (iv) heart rate and many other parameters. Moreover, the implementation of these devices using the Internet-of-Things (IoT) paradigm with smart sensor networks and wireless communication protocols have led to the uprising of Internet-of-Medical-Things (IoMT). There are several technological challenges to face when dealing with IoMT systems, mainly because typically these devices (i.e., WHD nodes) are battery powered and have to meet strict energy consumption and size requirements. Furthermore, the presence of a large number of WHD nodes in a network results in a significant quantity of data to be acquired, transmitted and stored. In order to deal with these challenges, the state of the art proposes data compression techniques, so as to reduce the amount of acquired data that have to be transmitted to the sink node. In fact, data transmission represents the most expensive activity in terms of energy consumption for an IoMT node [1]. In particular, the transmission of the ECG signals is one of the most power demanding activities. Typically, the compression of ECG signals is based on Domain Transform Methods (DTMs) [2,3,4], which ensure almost no information loss of the clinical information of the patient, but these methods require a high computational loads at the WHD nodes, which translates to a higher power consumption. In order to overcome this bottleneck, lossy compression methods based on Compressed Sensing (CS) have been developed and proposed in the literature to be used on WHDs. In particular, digital CS methods require low computational loads in the compression step of ECG signals, and a higher one in the reconstruction step. For these reasons, the compression step can be performed directly on WHD by means of CS, while the latter can be achieved by the receiving node that has enough computation power, without energy consumption constraints (e.g., PC, server). In [5], the authors compare a CS-based method for energy-efficient real-time ECG compression with several state of the art Discrete Wavelet Transform (DWT)-based compression methods. The authors present results which demonstrate that, even if the reconstruction quality of CS-based compression is lower than the classical DWT methods, it ensures a much higher energy efficiency and a lower computational load at the sink node, making it suitable for real-time ECG compression and decompression for remote healthcare services. In [6], the authors compared two lossy compression methods: CS compression vs. the Set Partitioning In Hierarchical Trees (SPIHT) algorithm. The performance of CS is lower when compared to SPIHT, but the CS-based method proposed in [6] exhibits lower distortion on the reconstructed ECG signal and lower power consumption, making it very appealing for applications with tight constraints such as those required for WHDs.

An important aspect is that the reconstruction quality of CS methods depends on how the sensing matrix and the dictionary matrix are chosen. Typically, random matrices based on well-known probability functions, such as Gaussian, Bernoulli, or Rademacher, are used as sensing matrices [6,7,8,9].

Alternatively to random matrices, which require the generation of random numbers in the WHD, recently deterministic matrices have been also proposed with performances in some cases better than random matrices. Regarding dictionary matrices, they are used for the reconstruction of the signal by the receiver node. The most utilized ones are based on domain transformation such as: Discrete Cosine Transform (DCT), Discrete Wavelet Transform (DWT) and Discrete Fourier Transform (DFT) [10,11].

Lately, in order to improve the reconstruction performance of the CS-based ECG signal acquisition, several authors explored the possibility of using a dictionary matrix trained and adapted to a specific patient. This approach is defined in the literature as Dictionary Learning (DL) and it aims to find a proper representation of datasets by means of reduced dimension dictionaries, adapted to the patient-specific ECG signal morphology [12]. In [13], the performance improvement due to the application of dictionary learning techniques to ECG signals is described. The authors compare the performance of an agnostic learned dictionary and a patient-specific learned dictionary, with two untrained dictionaries. The results show that DL, in particular when tuned on a single patient, helps to improve, considerably, the reconstruction quality of the ECG signal. Similarly, the advantages of DL are also demonstrated in [14], where the authors propose an ECG compression method where the dictionary contains segments of the ECG signal of the patient. Typically, DL techniques are applied to overcomplete dictionaries; this is due to the fact that the subspaces used for data description are usually greater than the signal dimension. This means that the observed processes are sparse in the set of all possible cases. So, the goal of DL is to find these subspaces in order to efficiently represent the compressed signals. The state of the art of DL methods can be divided into three major groups [12]: (i) the probabilistic learning methods, (ii) the learning methods based on clustering or vector quantization, and (iii) the learning methods for dictionaries with particular construction, typically driven by a priori knowledge of the signal structure. The work presented in [15] falls into the first group. Here, the authors use the Method of Optimal Direction (MOD) in order to train the dictionary matrix. The most representative learning algorithm belonging to the second group is instead the K-SVD. In [16], the authors present a method for enhancing the spatial resolution of the ECG signal by using a joint learning approach, based on the K-SVD algorithm of two patient’s dictionaries, one for the High Resolution (HR) ECG and one for the Low Resolution (LR) ECG. In [17], again the K-SVD algorithm is used. Moreover, in the reconstruction stage, a coarse signal is obtained using a generic dictionary. This coarse signal is used to detect the position of the QRS complex. By considering, again, the QRS position, it is possible to chose a new dictionary, in order to properly reconstruct the ECG signal. This new dictionary has been trained with signals having the same QRS complex position of the signal to reconstruct. In this way, the authors demonstrate that it is possible to increase the performance of standard dictionary learning-based CS methods.

The main difference between the methods in the first two groups and those in the last group is that probabilistic and clustering learning methods follow a fully data driven approach, by constructing the dictionary by a portion of the ECG signals acquired from the sensors. The alternative approach of the methods in the third group instead relies on parametric dictionaries, where dictionaries are built from a generating function and a set of atom parameters.

In practical applications, this approach is quite beneficial in terms of memory requirements, communication costs and implementation complexity [12]. This paper proposes a new method belonging to the third group among those mentioned above, which relies on a parametric dictionary. In particular, the method starts from a large wavelet parametric dictionary which is then reduced by means of a learning phase. Thanks to the learning phase, the dictionary adapts to the patient-specific features. The reduced dictionary is used for the signal reconstruction.

The paper presents the following innovations:It makes use of a dynamic sensing matrix, proposed in [18], which already proved to have superior performance compared with random and deterministic matrices;It performs the reconstruction on signals frames where the R peaks are aligned. Thanks to this alignment, the dictionary optimization can exploit the different scaling features of the ECG waves, and thus provide a waveform better matched to the ECG wave lying in a specific position. The optimization of the dictionary considering the waveform position was already exploited in [17]. In said paper, however, different dictionaries are created according to different positions of the QRS complex. In this paper, instead, the R-peak alignment of the frames during the acquisition phase is proposed.It is worth noting that modern state of the art ECG front-end chips for wearable devices, such as [19], are able to provide the position of the R peak together with the acquired samples. This means that the R-peak detection does not require computational load at the micro-controller side.It proposes a procedure for the dictionary optimization by modeling the optimization as a multiple measurement vector problem;It evaluates three different large wavelet dictionaries to be optimized and three different algorithms to solve the matrix optimization. In order to achieve a better performance, a parametric evaluation for the algorithms is carried out, since these parameters directly affect the estimation of the optimized matrix.

The proposed method and an initial evaluation of its performance are presented in [20]. In this paper, a more complete evaluation on ECG signals from the MIT-BIH Arrhythmia Database [21] is presented, and a comparison with other methods using dictionary learning is reported.

The proposed method has been designed as a solution for ECG monitoring, to be deployed on low power devices, such as the smart WHD of the Ambient-intelligent Tele-monitoring and Telemetry for Incepting & Catering over hUman Sustainability (ATTICUS) project [22], aiming at developing a telemedicine system.

The rest of the paper is organized as follows. In Section 2, an overview on CS is given. In Section 3, the proposed method and the algorithms evaluated for the dictionary optimization are presented, followed by the experimental evaluation in Section 4. Lastly, in Section 5, conclusions and future work are reported.

## 2. Compressed Sensing Overview

Compressed Sensing represents a novel framework for compressed acquisition and reconstruction of natural (or man-made) signals which present sparsity in a specific domain in which they can be represented. The aim of CS is to acquire a compressed version of the signal of interest and reconstruct it from this compressed acquisition. Let $x\in R^{N\times 1}$ be the vector of *N* samples acquired at Nyquist rate and $y\in R^{M\times 1}$ be its compressed acquisition; then, the CS compression process can be described as:(1)y=Φ·x
where M≪N, and $\Phi \in R^{M\times N}$ is the *sensing matrix*. In order to reconstruct the signal from its compressed acquisition, x must have a sparse representation in a specific domain (i.e., the signal can be represented by few coefficients in the chosen domain) [23,24]. If this condition is satisfied, it is in fact possible to reconstruct the signal from a relatively small number of samples. The ECG signal can be represented with a sparse signal model [8], and so it is suitable to be compressed using a CS approach. The sparse representation of the ECG signal can be modeled as: (2)x=Ψ·α
where $\Psi \in R^{N\times P}$ is the dictionary matrix and $\alpha \in R^{P\times 1}$ is the coefficient vector of the signal *x* in the transform domain, with *P* being the number of waveforms in the dictionary. Substituting (Equation 2) into (Equation 1), the following expression holds: (3)y=Φ·x=Φ·Ψ·α.

The reconstruction problem must take the *M* measurements in the vector y, the sensing matrix Φ and the dictionary matrix Ψ, and reconstruct the signal x. Since M≪N, there are infinite solutions to (Equation 3). Under the assumption that the vector of the signal coefficients α is *K*-sparse, it has been demonstrated that an estimation of the coefficient α can be obtained by solving: (4)$$\hat{\alpha }=\arg\min_{\alpha }{∥\alpha ∥}_{0}\;\mathrm{subject}\;\mathrm{to}:\;y=\Phi \;\cdot \;\Psi \;\cdot \;\alpha .$$
where ${∥\;\cdot \;∥}_{0}$ represents the $\ell_{0}$ norm operator. The minimum value of *M*, allowing a successful signal reconstruction, depends on the sparsity *K* and on the coherence between the sensing and the dictionary matrices [23]. Equation (Equation 4) represents a constrained optimization problem, where the aim is to find the vector α as the *maximally sparse* solution, subject to (Equation 3). However, since the positions of the nonzero elements in the vector are not known, the problem has a combinatorial complexity. For this reason, the (Equation 4) is often relaxed to a $\ell_{1}$ optimization problem [25], which instead can be solved by linear programming, as:(5)$$\hat{\alpha }=\arg\min_{\alpha }{∥\alpha ∥}_{1}\;\mathrm{subject}\;\mathrm{to}:\;y=\Phi \;\cdot \;\Psi \;\cdot \;\alpha .$$

Once the optimization in (Equation 4) or (Equation 5) has been solved, and the vector $\hat{\alpha }$ has been found, it is possible to reconstruct the ECG signal from the compressed samples using (Equation 2).

## 3. Proposed Method

The proposed method is reported in Figure 1 and consists of two phases: an initial *Training Phase* (Figure 2), which allows one to build the optimized dictionary, and an *ECG monitoring phase* (Figure 3), where the acquired ECG is compressed and the reconstruction can rely on the optimized dictionary. Considering a reference IoMT architecture, as described in [22], a brief explanation is presented.

The architecture consists of three main elements: (i) the smart wearable device, (ii) the server, and (iii) the end user. These three elements are located on the three layers depicted in Figure 1, respectively, as: (i) physical layer, (ii) information integration layer, and (iii) application service layer. The smart wearable device consists of a piece of clothing with several sensors embedded in order to acquire the ECG signal of the subject. Its tasks are: (i) to acquire the ECG signal, (ii) to split the acquired signal into frame, aligned to the R Peak, and (iii) to transmit the compressed (or uncompressed in the case of the training phase) ECG samples via a wireless communication interface to the server. The server receives the ECG signal and the unoptimized dictionary matrix during the training phase in order to realize the matrix optimization and store the optimized dictionary matrix that will be used in the *ECG monitoring phase*. During the *ECG monitoring phase*, the server will transmit both the signal received by the smart wearable device and the optimized dictionary matrix to the end user. The end user will then reconstruct the original signal.

### 3.1. Training Phase

A block scheme of the *Training Phase* is presented in Figure 2. In this phase, the proposed dictionary matrix optimization method is performed. For this purpose, several ECG signals sampled at the Nyquist rate are utilized, split into frames, one for each heart beat, and aligned with reference to the R peak. In particular, for each frame, a new record larger than the longest heartbeat is created and the samples of the frame are copied in the new record such that the sample corresponding to the R peak lies at a fixed percentage of the record.

Then, the aligned records are used for the dictionary optimization. The optimization stage, receives as input an unoptimized dictionary matrix and selects the columns of this matrix. The selection is carried out by taking into account the coefficients that better represent the signals in the domain defined by the dictionary matrix. As a result of this block, an optimized matrix is generated composed by the selected columns. For this phase, three algorithms have been evaluated. The details of the algorithms used are described in Section 3.3.1.

In the *Training phase*, the samples are acquired by the sensor nodes, disposed at the physical layer. During this phase, the sensor nodes send the uncompressed samples to the *Information Integration Layer*, which will realize the matrix optimization.

This *Training Phase* has a double advantage:it allows to select the dictionary columns that best match the ECG signal of the patient, thus improving the reconstruction quality. If the training set is properly chosen including a sufficient number of anomalous beats, the dictionary optimization should also provide a better reconstruction quality of the anomalous beats,it allows reducing the processing time of the ECG signal reconstruction, as the computational complexity of the algorithms increases with the number of columns of the dictionary matrix.

### 3.2. ECG Monitoring Phase

In the *ECG Monitoring Phase*, the optimized dictionary is used to reconstruct the ECG waveform from the compressed samples. A block scheme of this phase is reported in Figure 3.

Similarly to the previous phase, the Nyquist-sampled ECG signals are acquired, split into frames and aligned with reference to the R peak. Then, the CS encoding is performed by means of the sensing matrix Φ, and the compressed vector y is obtained. The sensing matrix is built dynamically, according to the algorithm described in [18], and briefly reported in Section 3.4.1.

In the reconstruction stage, the optimized dictionary has been used to recover the original waveform. With reference to the implementation, the compressed vectors are acquired by the sensor nodes at the physical layer and sent to the *Information integration layer*. Here, when requested, the waveforms are reconstructed using the optimized dictionary and eventually sent to the end user of the *Application Service Layer*.

### 3.3. Adopted Unoptimized Dictionary Matrices

In this paper, the unoptimized dictionary Ψ is chosen according to the analysis presented in [24], where the authors compared several dictionaries for ECG compression and reconstruction. The authors found that the best reconstruction performance is achieved by a dictionary defined according to the Mexican Hat kernel expressed as follows:(6)$$\psi \left(a,b\right)=\frac{2}{\sqrt{3a}\;\cdot \;\pi^{1/4}}\;\cdot \;\left[1-{\left(\frac{n-b}{a}\right)}^{2}\right]\;\cdot \;\exp^{-\frac{1}{2}{\left(\frac{n-b}{a}\right)}^{2}}$$
where n=0,…,N−1, *a* is the scaling factor of the Mexican Hat kernel and *b* is the delay factor. Three dictionaries are utilized in this paper, built by choosing different values of the scaling and delay factors. A record of ECG signal is acquired for several minutes (at least two) by the sensors and split into *F* frames, one for each heartbeat and are then aligned on the R-peak.

By considering all the frames of the training set, (Equation 2) can be rewritten as:(7)X=Ψ·A
where $X\in R^{N\times F}$ is a matrix built by placing the training records divided into F frames placed side-by-side columns-wise and $A\in R^{P\times F}$ is the matrix which contains the coefficients for the training records. It is worth noting that the higher the number of columns of Ψ, the higher the number of coefficients representing a single record is.

The idea underlying the proposed method is that, since the dictionary is large, and also that the frames are aligned to the R peak, several elements of the dictionary Ψ can be discarded, and instead only the most representative elements should be considered in reconstructing the signal. Based on this observation, the dictionary optimization problem can be written as the following problem, called Multiple Measurement Vector (MMV) sparse recovery:(8)$$I=\arg\min_{A}\left|\mathrm{supp}\left(A\right)\right|\;\mathrm{subject}\;\mathrm{to}:\;X=\Psi A$$
where given a matrix Z, supp(Z) is the *support* of Z, which is the index set of the rows which contain nonzero entries [26]. The solution of the above problem is a set of row indexes of the matrix A, corresponding to the most significant coefficients of the dictionary which are present in the training signals. Therefore, an optimization of the dictionary matrix Ψ can be carried out by selecting the columns of Ψ whose indexes are in the set I.

To solve the MMV sparse recovery problem in (Equation 8), three algorithms have been evaluated: Multiple Orthogonal Matching Pursuit (M-OMP), Multiple FOCal Underdetermined System Solver (M-FOCUSS) and Spectral Projected Gradient for Least square 1 (SPGL1). All the considered algorithms present a convergence or regularization parameter to be set in order to achieve a proper dictionary optimization and it is difficult to determine a priori the best value of such parameter according to the noise characteristics of the signals.

Therefore, in the *Training phase*, the matrix optimization is executed several times, for different values of the parameters, in a given range, and its performance is evaluated on the training signals, in terms of PRD. Among the obtained matrices (one for each value), the matrix $\Psi_{\alpha }$ that achieves the lowest PRD is selected to be used in the *ECG Monitoring Phase*. Details about the used algorithms and about their utilization for the matrix optimization will be provided in Section 3.3.1 and Section 4.1, respectively.

The proposed dictionary optimization method in this paper was evaluated on three different unoptimized matrices, all based on the Mexican Hat kernel:$\Psi_{1}$ is a dyadic matrix, defined according to [18], and expressed by (Equation 9). Here, the scaling factor *a* follows the power of 2 (i.e., a=2 for the first $\frac{N}{2}$ values, a=4 for the following $\frac{N}{4}$, and so on), while the delay factor *b* varies from 0 to $\frac{\left(N-1\right)}{a}$ with a step of *a*;$\Psi_{2}$ is a Mexican Hat matrix where *a* also follows the power of 2, while *b* varies linearly from 0 to N−1 with a unitary step (see (Equation 10));$\Psi_{3}$ is a matrix (see (Equation 11)) where *a* follows the geometric progression $2^{n}$, $n\in [1,\log_{2}N]$ with a step of $\frac{1}{2}$, and *b* varies from 0 to $\frac{\left(N-1\right)}{a}$ with a step of *a*.
(9)$$\begin{array}{c} \Psi_{1}=[\psi \left(2,0\right),\psi \left(2,2\right),\psi \left(2,4\right),\ldots ,\psi \left(2,2\left[\frac{N-1}{2}\right]\right),\\ \psi \left(4,0\right),\psi \left(4,4\right),\psi \left(4,8\right),\ldots ,\psi \left(4,4\left[\frac{N-1}{4}\right]\right)\\ \ldots ,\psi (N,0)]. \end{array}$$
(10)$$\begin{array}{c} \Psi_{2}=[\psi \left(2,0\right),\psi \left(2,1\right),\psi \left(2,2\right),\ldots ,\psi \left(2,\left[N-1\right]\right),\\ \psi \left(4,0\right),\psi \left(4,1\right),\psi \left(4,2\right),\ldots ,\psi \left(4,\left[N-1\right]\right)\ldots ,\\ \psi (N,N-1)]. \end{array}$$
(11)$$\begin{array}{c} \Psi_{3}=[\psi \left(2,0\right),\psi \left(2,2\right),\psi \left(2,4\right),\ldots ,\psi \left(2,2\left[\frac{N-1}{2}\right]\right),\\ \psi \left(2\sqrt{2},0\right),\psi \left(2\sqrt{2},2\sqrt{2}\right),\psi \left(2\sqrt{2},4\sqrt{2}\right),\\ \ldots ,\psi \left(2\sqrt{2},2\sqrt{2}\left[\frac{N-1}{2\sqrt{2}}\right]\right)\\ \ldots ,\psi (N,0)]. \end{array}$$

For all the considered matrices, an additional column $u={\left[1/N,\ldots ,1/N\right]}^{T}\in R^{N\times 1}$ has been added in order to take into account possible biases of the ECG signals (e.g., the baseline wander).

#### 3.3.1. Dictionary Optimization Algorithms

In this subsection, the algorithms used for solving the MMV sparse recovery problem (Equation 8) are briefly recalled.*M-OMP*

This algorithm falls into the forward sequential selection methods, aiming to find a sparse solution by sequentially building up a smaller subset of column vectors from the dictionary matrix in order to represent the signal X [27]. The algorithm performs the following steps [27,28]:The residual $R_{0}=X$, the set of column indices ***Λ*** = ∅ and the counter iteration *t* = 1 are initialized;Find the index $\lambda_{t}$ that solves the equation
(12)$$\lambda_{t}=\arg\max_{j\in [1,N]}\left(\frac{\left|\right|z_{j}{\left|\right|}_{2}}{\left|\right|\psi_{j}{\left|\right|}_{2}}\right),$$
where $z_{j}=R_{t-1}^{T}\psi_{j}$ and $\psi_{j}$ is the j-th column of Ψ;***Λ*** and Ψ are updated, $\Lambda_{t}=\Lambda_{t-1}\cup \lambda_{t}$, $\Psi_{t}=\left[\Psi_{t-1}\;\Psi_{\lambda_{t}}\right]$;The least square approximation is performed and the $\alpha_{t}$ vector is evaluated:
(13)$$\alpha_{t}=\arg\min_{\alpha }||X-\Psi_{t}\alpha {\left|\right|}_{2}.$$The new residual is calculated:
(14)$$R_{t}=X-\Psi_{t}\alpha_{t}.$$The iteration counter *t* is incremented, and the algorithm checks for the following stop conditions:
*t* is greater than the number of ECG samples per frame N;The new residual $R_{t}$ is lower than a fixed threshold $r_{th}$.If none of those two conditions are met, the algorithm returns to step 2.

The M-OMP algorithm is the quickest to optimize. This is due the fact that the algorithm returns the reached residual value at every iteration and the set of indices Λ utilized. In order to achieve the lowest possible PRD value, the following tasks are performed:Set the number of iterations of the M-OMP algorithm and run it;At the last iteration, the algorithm returns a vector of the residuals, each corresponding to an iteration;Each element $r_{k}$ of the residuals vector is associated to a subset $\Lambda_{k}$ of Λ;A range of residuals is selected, and accordingly, a set of candidate optimized matrices is built by selecting the corresponding subsets;Set the desired USR value;Run the OMP algorithm with all the possible optimized matrix obtained in step 4;Choose the optimized matrix that achieves the lowest PRD value.*M-FOCUSS*

The FOCUSS algorithm aims to find a solution for dictionary optimization that is referred in the literature as a weighted minimum norm solution. This solution is defined as the one that minimizes a weighted norm $\left|\right|W^{-1}\alpha {\left|\right|}_{2}$, where W is called weight matrix. The algorithm starts by finding a coarse solution for the representation of the sparse signal and at every iteration; this solution is pruned, by means of reducing the dictionary size. This process is implemented using a generalized Affine Scaling Transformation (ASL) in which the weight matrix, chosen initially as an identity matrix, is calculated at every iteration [29]. The solution is expressed by:(15)$$\alpha =W{\left(\Psi W\right)}^{+}x$$
where $W\in R^{P\times N}$ and ${\left(\;\cdot \;\right)}^{+}$ denotes the Moore–Penrose inverse.

In this work, the utilized FOCUSS algorithm is called regularized FOCUSS because it includes a regularization method called Tikhonov regularization, which is based on the inclusion of a regularization parameter, λ. Furthermore, it is assumed that multiple signals are acquired that share a similar sparsity profile and dictionary. Under these assumptions, the model can be posed as an MMV problem as follows:(16)$$\arg\min_{A}{\;∥\Psi A-Y∥}_{F}^{2}+\lambda \sum_{i=1}^{N}(∥\alpha_{i}{{∥}_{2})}^{p}$$
where ${∥.∥}_{F}$ is the Frobenius norm. This version of the algorithm is commonly referred to as M-FOCUSS. At every iteration, the algorithm builds a dictionary matrix with a reduced number of columns with respect to the one obtained from the previous iteration. Since λ is chosen as a constant value, the tuning process performed for the M-FOCUSS algorithm is based on the number of iterations and is synthesized in the following steps:Set the desired USR value, a large number of iterations and a small value of the λ parameter as reported in [Equation 26]. At each iteration, the algorithm returns a progressively smaller dictionary matrix. These matrices are used as a possible optimized dictionary matrix;Reconstruct the ECG signals with all the possible dictionary matrices obtained at each iteration of M-FOCUSS;Choose the optimized matrix that achieves the lowest PRD value.*SPGL1*

The SPGL1 algorithm is described in detail in [30]. The algorithm finds the solution of MMV Basis Pursuit (BP) problem by iterative solving an associated MMV Least Square (LS) problem. The BP problem can be expressed as
(17)$$\arg\min_{A}{\;||A||}_{1}\;\mathrm{subject}\;\mathrm{to}:\;||\Psi A-{X||}_{F}^{2}\leq \sigma$$

In order to obtain a sparse solution, the BP problem is redefined as an LS problem:(18)$$\arg\min_{A}\;||\Psi A-{X||}_{F}^{2}{\;\mathrm{subject}\;\mathrm{to}:\;||A||}_{1}\leq \tau .$$
τ and σ are positive parameters that can be seen as an estimation of the noise level in the data or the error from the ideal BP solution and LS one, where σ and τ are 0. If the τ parameter is set appropriately, namely, τ is equal to $\tau_{\sigma }$, the solution of the BP problem and the LS problem is the same. Choosing the parameter σ, the algorithm implements a Newton-based method in order to update τ to achieve the value $\tau_{\sigma }$, where $\tau_{\sigma }$ is the value of the parameter where the BP solutions and the LS solutions coincide. Once the optimal value of the τ parameter is found, and the LS solution is equal to the BP solution, a spectral projected-gradient is used to solve (Equation 18).

The SPGL1 takes as an input the desired value of σ, namely $\sigma^{*}$, and returns at each iteration the σ value reached. At the *i*-th iteration, this value will be expressed as $\sigma_{i}$. Since this parameter can be seen as an estimation of the noise level in the data, $\sigma^{*}$ must be chosen properly: if the value is too low, the noise level is underestimated, while if the value of $\sigma^{*}$ is too high, the noise level is overestimated. In the first case, $\sigma_{i}$ will never reach the objective value $\sigma^{*}$, while in the second case, the optimized dictionary is made with too few columns and cannot assure the reconstruction of the signal. In order to assign the correct values to σ, and proceed with the performance evaluation for the tuning, the following steps are performed:Set the desired USR value, a large number of iterations (e.g., 1000) and the lowest $\sigma^{*}$ value among those taken into consideration in this work (i.e., $\sigma^{*}$ = 0.2). Now two scenarios are possible: (i) $\sigma_{i}=\sigma^{*}$ at the *i*-th iteration, so the first value chosen for $\sigma^{*}$ will produce an optimized dictionary matrix and all the other values of $\sigma^{*}$ can be used as reported in (Equation 25); (ii) $\sigma_{i}$ will not reach the objective value $\sigma^{*}$, but $\sigma_{i}$ is an output of the algorithm. The value of $\sigma^{*}$ is updated as the nearest upper value presented in (Equation 25) with respect to the output $\sigma_{i}$. Step 1 is repeated again until condition (i) is satisfied;Since the lowest value of $\sigma^{*}$ has been found from step 1, starting from this value, the algorithm runs for all the other $\sigma^{*}$ values reported in (Equation 25). For each of these values, an optimized matrix is returned from the algorithm;Reconstruct the ECG signal utilizing all the optimized matrices obtained from the previous step;Choose the optimized matrix that achieves the lowest PRD value.

### 3.4. ECG Monitoring Phase

In this section, the compression and the reconstruction steps of the ECG signals by means of the CS method proposed in this work are presented.

#### 3.4.1. ECG Signal Compression by Means of CS

In the scientific literature, when dealing with the application of CS for ECG monitoring, often random sensing matrices, based on Bernoulli or Gaussian random probability functions, are used. The performance of these random matrices heavily depends on the correlation between the sensing matrix elements and the acquired samples. The compression algorithm presented in [18] overcomes this limitation by adopting a deterministic sensing matrix that depends on the ECG signal to be compressed. By constructing a sensing matrix adapted to the ECG signal, it contains more information about the signal features, and therefore exhibits a better reconstruction performance.

The algorithm described in [18], creates a sensing matrix Φ which is chosen in a such way that y represents a sort of auto-correlation of the signal x, containing the ECG samples at the Nyquist rate. More precisely, the compressed vector y is obtained as the cross-correlation of x and a binary vector p, whose elements are 1 if the magnitude of the corresponding sample in x is above a specified threshold and 0 otherwise. The method operates on frames of *N* samples and for each record x, an average operation is performed, obtaining $x_{avg}$. The average is used for the evaluation of the magnitude $x_{a}$, as:(19)$$x_{a}=\left|x-x_{avg}\right|$$

For each frame, the magnitude is then compared with a threshold value $x_{th}$, which represents a certain percentile of the waveform amplitude. $x_{th}$ is evaluated by means of a sorting-based algorithm. When an update of the sensing matrix is needed, the *N*-size binary vector p is constructed by comparing the signal magnitude $x_{a}$ with the signal threshold $x_{th}$. Hence, the *n*-element of p is evaluated as:(20)$$p\left(n\right)=\left\{\begin{array}{cc} 1, & \mathrm{if}x_{a}\left(n\right)\geq x_{th}\\ 0, & \mathrm{if}x_{a}\left(n\right)\leq x_{th} \end{array}\right.,$$

The vector p is the first row of the sensing matrix Φ, whereas the other rows are a circular shifted version of the p vector by an integer quantity equal to the *Under Sampling Ratio* (*USR*), where USR=N/M. If a significant change in $x_{th}$ is found, the sensing matrix Φ is updated; otherwise, the sensing matrix of the previous frame is used. In order to be considered as a significant change, the distance between the threshold value at the current frame and the threshold value at the previous frame must be higher than a specified limit ε.

In all the frames where the sensing matrix is changed, the vector p is sent together with the compressed samples.

#### 3.4.2. ECG Signal Reconstruction by Means of CS

By knowing both the sensing matrix and the optimized dictionary, the OMP algorithm is used to estimate the vector $\hat{\alpha }$ using (Equation 4). Afterwards, the ECG waveform is reconstructed by using the following formula:(21)$$\hat{x}=\Psi_{\alpha }\;\cdot \;\hat{\alpha }$$

## 4. Experimental Results

The proposed method has been evaluated experimentally in the MATLAB environment on ECG signals taken from the PhysioNet MIT-BIH Arrhythmia Database [21]. The MIT-BIH Database has been chosen because it is the most utilized in the literature among the ECG databases available online. This database contains 48 half-hour excerpts of two-channel ambulatory ECG recordings, obtained from 47 subjects studied by the BIH Arrhythmia Laboratory. The recordings have been acquired with a sampling frequency of 360 Hz and a resolution of 11 bit per channel. The signals have been divided into frames with size N=512 and a filtering stage is applied on them in order to remove the first three harmonics of the power line signal (60, 120, 180 Hz). For the tests, ten ECG datasets have been used S={102,103,105,107,122,100,101,106,112,113} which contain the following beat labels: paced beats, normal beats, premature ventricular contraction, fusion of paced and normal beats, fusion of ventricular and normal beats and atrial premature beats. For M-OMP and SPGL1, the unoptimized dictionary matrices utilized are the one expressed in (Equation 9)–(Equation 11), while in the case of M-FOCUSS, only (Equation 9) has been considered, due to the poor performances exhibited by this algorithm while using (Equation 10) and (Equation 11). The optimization algorithms and the corresponding dictionaries have been tested using 10 min of ECG signals from the set S with different values of USR. For each dataset, the first 5 min were used in the *Training Phase*, while the remaining 5 min were used in the acquisition phase. By checking the annotation of every heartbeat, the signals were acquired by making sure that abnormal beats were present inside both phases. The reconstruction performance has been evaluated by means of the PRD as a figure of merit which is commonly adopted in the literature. The PRD is calculated for each frame, as follows:(22)$$\mathrm{PRD}=\frac{∥x-\hat{x}{∥}_{2}}{{∥x∥}_{2}}\;\cdot \;100\%$$
where x is the acquired ECG signal at Nyquist rate, without being compressed, and $\hat{x}$ is the reconstructed signal by means of CS according to Section 3.4.1. Then, the average PRD on the entire ECG signal has been obtained by taking the average of the PRD values calculated for each frame. After assessing the performance of the proposed method and algorithms, a comparison has been carried out with other dictionary learning methods, namely [5,14,15]. For this purpose, the Normalized PRD (PRDN) of the MIT-BIH ECG signal labeled No. 117 was evaluated and compared with the results expressed in [14] obtained using the same signal. The PRDN was defined as follows:(23)$$PRDN=\frac{∥x-\hat{x}{∥}_{2}}{∥x-x_{avg}{∥}_{2}}\;\cdot \;100\%$$

Compared to PRD, PRDN removes the average of the original signal, and thus it is not affected by the dc bias that could be present in some considered signals.

### 4.1. Convergence Parameters

All the three algorithms used in this work present a convergence parameter to be set, in order to achieve a proper optimization of the dictionary matrix. These parameters are:Residual threshold, $r_{th}$ for the M-OMP;σ, an estimation of the noise level in the data for the SPGL1;λ, a regularization constant for the M-FOCUSS.

These parameters directly affect the performance of the algorithms as expressed in (Equation 14), (Equation 16) and (Equation 17), where it is possible to note that they represent the constraints of the problems. A lower value of these parameters means a more strict constraint which is translated into an optimized matrix with an higher number of columns, while with higher values, the algorithms solve a more relaxed version of the problem. Both extremes are not desirable, the first due to the OMP algorithm in reconstruction, that has to work on a larger domain, degrading its performance, and the latter because it provides an optimized matrix with too few columns, not assuring the reconstruction of the compressed ECG signal. The optimal value to assign depends on the ECG signals that are used in the *Training Phase*. So, in order to improve the dictionary optimization, the *Training Phase* includes a parameter tuning stage. For the *Training Phase*, 5 min of ECG are acquired from the sensors, split into frame and aligned to the R-peak. A set of values of the parameters reported in Table 1 are chosen:(24)$$r_{th}=\left\{0.12,0.16,\ldots ,0.56,0.60\right\}.$$
(25)σ={0.2,0.3,…,1.9,2.0}.
(26)λ=0.00025,iterations=500.

The corresponding optimization algorithm is executed for each chosen value, and for each of them, a different optimized matrix is obtained. In order to evaluate the contribution of the parameters, the reconstruction stage is executed on all the optimized matrices and the PRD is calculated. The λ parameter of M-FOCUSS was chosen as a constant value. This is due to the fact that in order to find a solution to the optimization problem by utilizing the M-FOCUSS algorithm, the values of the regularization parameter λ should be found at every iteration of the algorithm, as stated in [27]. Although there are some methods that allow one to choose the values of λ to be used for every iteration ([27]), this leads to a noncomputationally efficient approach [29]. λ allows a balance between optimization quality (that translates directly into signal estimation quality) and sparsity (which leads to a faster reconstruction) [31]. For the tuning of the M-FOCUSS algorithm, in this paper, a very small constant value of λ has been chosen, so as to drive the algorithm towards a fine solution. Then, the performance of the reconstruction is analyzed at each iteration and the matrix giving the lowest PRD is selected.

### 4.2. Performance Evaluation

The experimental results for each dictionary and algorithm taken into account are shown from Table 2, Table 3, Table 4, Table 5, Table 6, Table 7, Table 8, Table 9, Table 10 and Table 11. The tests were performed after the tuning phase described in the previous subsection. In Figure 4, Figure 5 and Figure 6, an example of the improvement in terms of *PRD* for all three algorithms, used for the training of the dictionary matrix on the signal No. 122, is presented. In particular, in Figure 4, PRD values achieved by the dictionaries that have been provided as output at each iteration of the M-FOCUSS algorithm, are presented. Only the results of 150 iterations are shown instead of the nominal 500 because, in this case, the dictionaries obtained after 150 iterations do not assure a proper reconstruction of the signal. After reaching the minimum of the PRD, around iteration 100, the reconstruction error begins to grow again, due to the further reduction in the dictionary. In Figure 5 and Figure 6, not all the nominal values of $r_{th}$ and σ are shown, due to the fact that the dictionaries obtained with those values could not guarantee the signal recovery. It can be seen that by properly choosing the values of the regularization parameters, the reconstruction performances are enhanced.

The first column of all tables represents the USR value chosen for the compression of the ECG signal. The second column contains the PRD value, obtained using the unoptimized dictionary matrix expressed as (Equation 9) as in [18]. In the subsequent columns, the PRD values achieved using the matrices optimized by means of the proposed method are reported. In particular, $\Psi_{\alpha 1}$, $\Psi_{\alpha 2}$ and $\Psi_{\alpha 3}$ are obtained, starting from $\Psi_{1}$, $\Psi_{2}$ and $\Psi_{3}$, respectively. For USR values of 2 and 3, the performance obtained with the optimized matrices are comparable with that obtained with the unoptimized one. The situation changes when taking into account higher values of USR. A significant reduction in the PRD can be observed with the optimized dictionaries. By assessing the overall performance, the best results are achieved by $\Psi_{\alpha 2}$ followed by $\Psi_{\alpha 3}$ and $\Psi_{\alpha 1}$ for M-OMP and by $\Psi_{\alpha 2}$ for SPGL1 with M-FOCUSS right behind. In particular, the M-OMP exhibits the lowest values of PRD, with the matrix $\Psi_{\alpha 2}$ for signals Nos. 105 (Table 2), No. 107 (Table 5), No. 122 (Table 6), No. 100 (Table 7), No. 106 (Table 9) and No. 112 (Table 10), with the matrix $\Psi_{\alpha 3}$ for signals No. 103 (Table 3), No. 102 (Table 4), No. 101 (Table 8) and with the matrix $\Psi_{\alpha 1}$ for signals No. 113 (Table 11). The second lowest values of PRD are achieved by the SPGL1 with the matrix $\Psi_{\alpha 2}$ for the signal Nos. 103, 107, 122 and 100, with the matrix $\Psi_{\alpha 1}$ for signal Nos. 101 while for signal No. 112 the two matrices exhibit comparable results. Lastly, M-FOCUSS exhibits the second lowest values of PRD for signals Nos. 102, 106 and 113. By taking into account the PRD values achieved by the utilized algorithms, the best performance is obtained by deploying the M-OMP algorithm with the $\Psi_{2}$ dictionary for ECG signal No. 105, (Table 2) where it is possible to remain below the 9% threshold with an USR of 10, while the worst was obtained for ECG signal No. 101, where the maximum possible exploitable value of the USR is 6. In order to show the range of the obtained PRDs, Table 12 reports, for each *USR*, the lowest and highest obtained PRDs among the considered signals.

In Table 13, a comparison between the proposed method and the ECG compression techniques based on CS presented in [5,14] is reported. All the results have been obtained using the signal No. 117. It is possible to observe that the proposed method exhibits lower values of the PRDN for *USR*s lower than 8. The method in [14] achieves better results than the proposed one for *USR*s greater than 8. However, the obtained values of PRDN are high, such that the clinical content of the signal could be compromised. This makes such *USR*s not practically usable. In Figure 7, the best performance exhibited by the proposed method is compared with the one presented in [15]. The authors implemented a CS-based reconstruction method for ECG signals based on dictionary learning with the possibility of updating the matrix if the reconstruction error is too high. Even by considering this additional feature, the proposed method exhibits a better performance.

## 5. Conclusions

In this paper, a new approach for the optimization of a dictionary used in real-time ECG signal compression and reconstruction system based on CS was presented. The proposed approach is an alternative to fully data driven dictionary optimization methods, where the dictionary is constructed from the training data, and utilizes an overcomplete wavelet dictionary that is reduced by means of dictionary optimization algorithm, in order to leave only the highest impact columns for the purpose of the reconstruction of the ECG signals. Furthermore, in order to exploit the different scaling features of the ECG waves, an alignment of the frames according to the R-peak was used. Starting from an unoptimized dictionary matrix expressed by the Mexican Hat function using three different combination of its parameters, the proposed method includes a *Training Phase* where three different algorithms have been used for the dictionary optimization: M-OMP, SPGL1 and M-FOCUSS. The training phase also includes an experimental tuning of the convergence parameter of each algorithm, in order to ensure the most suitable condition for the optimization of the dictionary and the reconstruction of the ECG signals. The paper presented highlights the advantages of using an optimized dictionary matrix, assesses the influence of convergence parameters on the algorithms, and, therefore, on the dictionary optimization, and evaluates the performances of the proposed method with different optimization algorithms by comparing the results of the proposed method with other state of the art methods. The optimization allows one to reach a much higher performance due to the elimination of redundant columns from the dictionary and by decreasing the domain in which the reconstruction OMP algorithm works. Moreover, the introduced tuning stage allows ensuring that each considered algorithm works in its most suitable conditions, thus providing the best reconstruction results. For the performance evaluation of each algorithm, several ECG signals from the PhysioNet MIT-BIH Arrhythmia Database were considered, evaluating the *PRD* for several *USR* values. The analysis demonstrated that: (i) even on abnormal beats, the utilization of an optimized dictionary matrix improves the reconstruction performance of the ECG signals, and (ii) this performance is further increased by the tuning of the convergence parameters, which is fundamental for a correct dictionary optimization. Furthermore, among the considered cases, the best performance on average was achieved by the M-OMP algorithm while using the $\Psi_{\alpha 2}$ dictionary. By comparing the performance of the proposed method with other state of the art methods, it was proven to outperform the other reconstruction methods based on dictionary learning and patient-specific dictionaries, for *USR*s lower than 8.

## Figures and Tables

**Figure 1 sensors-21-05282-f001:**
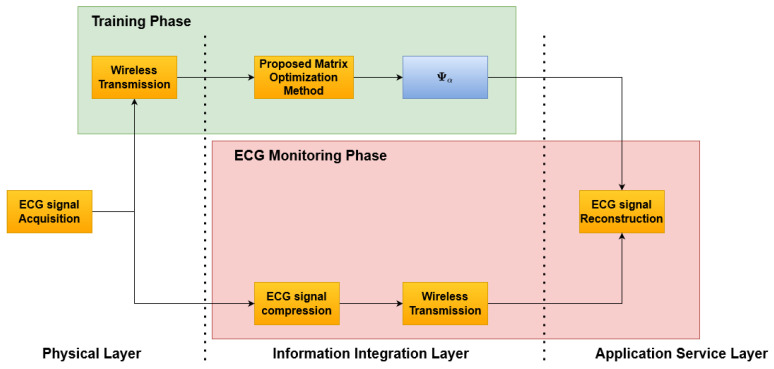
Block scheme of the Proposed Method.

**Figure 2 sensors-21-05282-f002:**
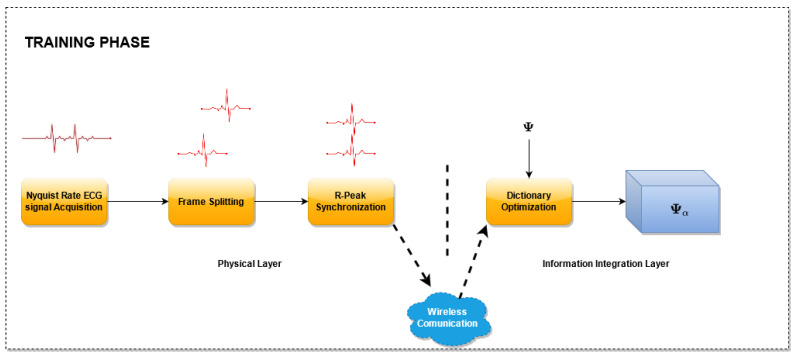
Block scheme of the training phase.

**Figure 3 sensors-21-05282-f003:**
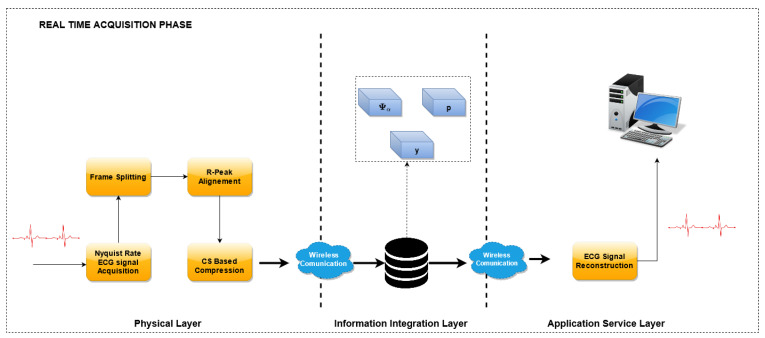
An overview of the utilization of the proposed dictionary matrix optimization process for continuous real-time ECG monitoring.

**Figure 4 sensors-21-05282-f004:**
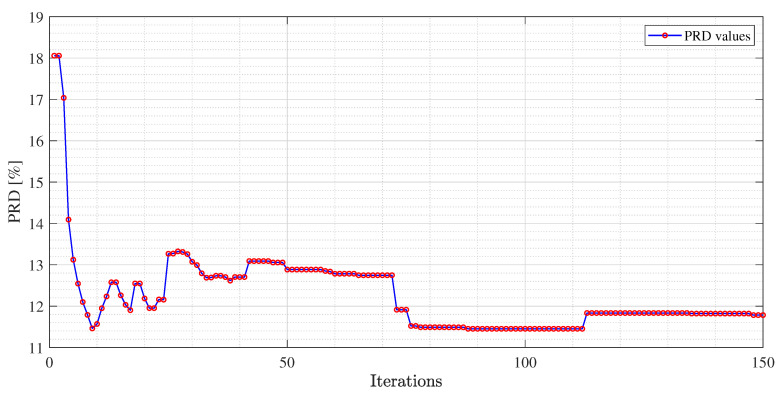
*PRD* values at every iteration of M-FOCUSS for *USR* = 9 for the MIT-BIH arrhythmia database signal No. 122.

**Figure 5 sensors-21-05282-f005:**
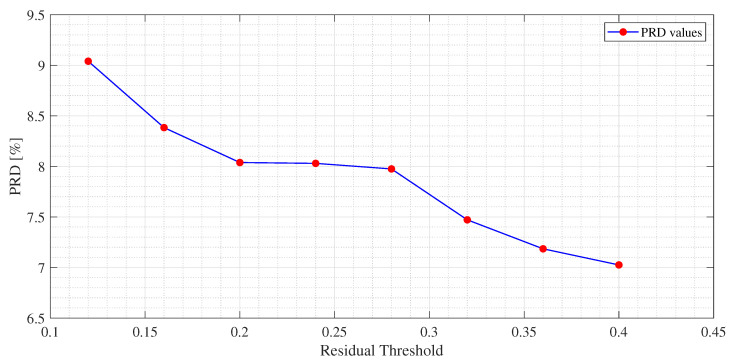
*PRD* values for several values of the Residual Threshold of the M-OMP for *USR* = 9 for the MIT-BIH arrhythmia database signal No. 122.

**Figure 6 sensors-21-05282-f006:**
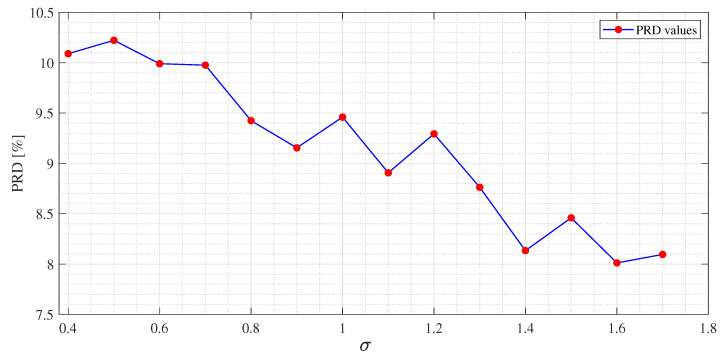
*PRD* values for several values of σ of SGPL1 for *USR* = 9 for the MIT-BIH arrhythmia database signal No. 122.

**Figure 7 sensors-21-05282-f007:**
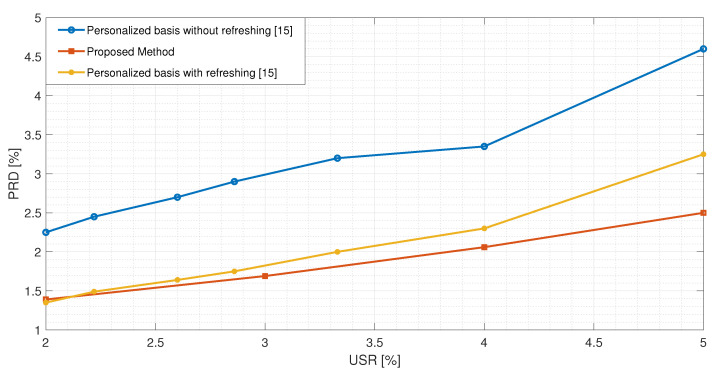
*PRD* value comparison between the proposed method and the method proposed in [15].

**Table 1 sensors-21-05282-t001:** Inputs, convergence parameter and output of the analyzed reconstruction algorithms.

Algorithm	Inputs	Parameter	Output
M-OMP	**X**, Ψ, iterations, $r_{th}$	$r_{th}$	**R**, Λ
M-FOCUSS	**X**, Ψ, λ, iterations	λ	$\Psi_{\alpha }$
SPGL1	**X**, Ψ, σ, iterations	σ	$\Psi_{\alpha }$, $\sigma_{t}$

**Table 2 sensors-21-05282-t002:** Comparison of PRD values from the algorithms using the three dictionary matrices at several *USR* values, for ECG signal no. 105.

		M-OMP			SPGL1			M-FOCUSS
***USR***	Ψ	$\Psi_{\alpha 1}$	$\Psi_{\alpha 2}$	$\Psi_{\alpha 3}$	$\Psi_{\alpha 1}$	$\Psi_{\alpha 2}$	$\Psi_{\alpha 3}$	$\Psi_{\alpha 1}$
2	1.69	1.54	1.39	1.38	1.45	1.32	1.37	1.43
3	2.10	1.95	1.69	1.65	1.87	1.61	1.68	1.66
4	3.24	2.49	2.06	2.12	2.56	2.13	2.24	2.20
5	5.02	3.21	2.50	2.60	3.44	2.78	2.99	3.01
6	7.12	4.23	3.21	3.15	4.94	3.59	4.01	4.11
7	10.29	5.26	3.54	3.86	6.31	4.65	5.19	5.06
8	12.59	6.26	4.11	4.56	7.85	5.66	7.09	6.18
9	17.59	8.27	5.58	5.98	10.36	7.61	9.83	7.92
10	20.47	12.20	7.84	8.02	13.58	10.25	13.01	10.23

**Table 3 sensors-21-05282-t003:** Comparison of PRD values from the algorithms using the three dictionary matrices at several *USR* values, for ECG signal No. 103.

		M-OMP			SPGL1			M-FOCUSS
***USR***	Ψ	$\Psi_{\alpha 1}$	$\Psi_{\alpha 2}$	$\Psi_{\alpha 3}$	$\Psi_{\alpha 1}$	$\Psi_{\alpha 2}$	$\Psi_{\alpha 3}$	$\Psi_{\alpha 1}$
2	1.25	1.17	1.13	1.18	0.99	0.86	0.91	0.93
3	2.15	2.70	1.52	1.69	1.43	1.23	1.36	1.28
4	3.50	3.18	1.99	2.18	2.20	1.97	2.53	2.09
5	6.79	4.41	2.54	2.66	3.62	3.40	4.28	3.02
6	10.37	7.50	3.45	3.45	7.02	4.96	6.63	4.66
7	14.14	8.71	4.31	4.85	9.34	6.11	8.46	6.00
8	17.12	11.38	4.95	6.24	12.67	6.59	11.62	6.89
9	24.14	17.16	6.41	7.07	17.80	9.85	18.36	8.85
10	32.78	18.97	9.60	8.73	21.82	14.59	24.91	15.73

**Table 4 sensors-21-05282-t004:** Comparison of PRD values from the algorithms using the three dictionary matrices at several *USR* values, for the ECG signal No. 102.

		M-OMP			SPGL1			M-FOCUSS
***USR***	Ψ	$\Psi_{\alpha 1}$	$\Psi_{\alpha 2}$	$\Psi_{\alpha 3}$	$\Psi_{\alpha 1}$	$\Psi_{\alpha 2}$	$\Psi_{\alpha 3}$	$\Psi_{\alpha 1}$
2	2.31	2.33	2.27	2.18	2.02	1.90	1.88	2.17
3	3.08	3.07	2.95	2.79	2.65	2.51	2.51	3.32
4	5.01	4.32	4.23	4.32	3.71	3.61	3.64	5.19
5	7.43	5.52	5.16	5.25	5.18	4.91	4.93	8.28
6	10.46	7.82	6.44	7.53	7.17	6.83	6.49	8.31
7	13.24	9.43	7.83	8.84	8.80	8.55	8.29	8.61
8	15.38	10.95	9.39	10.60	9.91	10.02	10.61	9.87
9	19.27	12.58	12.16	12.53	12.08	12.22	13.54	11.99
10	22.78	14.71	17.65	13.37	14.36	15.19	17.07	14.28

**Table 5 sensors-21-05282-t005:** Comparison of PRD values from the algorithms using the three dictionary matrices at several *USR* values, for  ECG signal No. 107.

		M-OMP			SPGL1			M-FOCUSS
***USR***	Ψ	$\Psi_{\alpha 1}$	$\Psi_{\alpha 2}$	$\Psi_{\alpha 3}$	$\Psi_{\alpha 1}$	$\Psi_{\alpha 2}$	$\Psi_{\alpha 3}$	$\Psi_{\alpha 1}$
2	0.81	0.82	0.76	0.73	0.77	0.87	0.72	0.78
3	1.23	1.28	1.23	1.32	1.14	1.30	1.01	1.20
4	2.25	2.45	2.03	3.29	1.87	1.67	1.67	1.67
5	3.73	3.58	3.29	4.28	2.43	2.31	2.60	2.24
6	5.42	4.77	4.36	5.03	3.88	3.45	3.67	3.64
7	7.89	6.40	5.26	5.61	5.86	4.68	5.16	5.57
8	10.43	9.37	6.27	8.66	7.68	5.93	6.95	7.43
9	13.13	15.52	7.77	13.67	11.31	7.90	8.83	10.55
10	15.49	18.76	9.72	17.06	14.22	10.33	10.94	13.35

**Table 6 sensors-21-05282-t006:** Comparison of PRD values from the algorithms using the three dictionary matrices at several *USR* values, for  ECG signal No. 122.

		M-OMP			SPGL1			M-FOCUSS
***USR***	Ψ	$\Psi_{\alpha 1}$	$\Psi_{\alpha 2}$	$\Psi_{\alpha 3}$	$\Psi_{\alpha 1}$	$\Psi_{\alpha 2}$	$\Psi_{\alpha 3}$	$\Psi_{\alpha 1}$
2	1.27	1.37	1.24	1.24	1.09	1.03	1.02	1.16
3	1.71	2.01	1.56	1.65	1.46	1.32	1.32	1.44
4	2.99	2.89	2.05	2.44	2.13	1.83	2.13	2.08
5	4.96	3.85	2.64	3.04	3.03	2.51	2.93	3.05
6	7.26	5.17	3.46	3.82	4.67	3.63	4.35	4.83
7	9.40	6.72	4.35	5.08	6.25	4.79	6.15	7.03
8	11.39	8.29	5.50	7.94	7.46	5.98	8.00	8.87
9	15.27	10.71	7.02	10.74	10.14	8.09	10.26	11.49
10	18.73	13.18	10.24	13.25	13.85	11.12	13.10	15.74

**Table 7 sensors-21-05282-t007:** Comparison of PRD values from the algorithms using the three dictionary matrices at several *USR* values, for  ECG signal No. 100.

		M-OMP			SPGL1			M-FOCUSS
***USR***	Ψ	$\Psi_{\alpha 1}$	$\Psi_{\alpha 2}$	$\Psi_{\alpha 3}$	$\Psi_{\alpha 1}$	$\Psi_{\alpha 2}$	$\Psi_{\alpha 3}$	$\Psi_{\alpha 1}$
2	2.73	2.94	2.49	2.44	2.33	2.00	1.89	2.73
3	3.71	4.37	3.15	3.13	3.00	2.64	2.58	3.70
4	5.86	5.47	4.21	4.08	4.01	3.58	3.14	4.92
5	10.24	6.87	5.33	5.27	5.26	4.82	4.23	6.54
6	15.15	10.58	6.67	6.92	7.47	6.90	6.16	8.15
7	20.74	13.16	7.57	8.18	9.72	8.19	8.62	10.67
8	23.99	15.75	8.65	9.55	11.37	9.67	11.62	12.31
9	31.70	20.37	9.77	11.22	14.42	12.47	14.36	16.40
10	38.22	21.91	12.69	15.90	18.90	16.20	18.25	21.25

**Table 8 sensors-21-05282-t008:** Comparison of PRD values from the algorithms using the three dictionary matrices at several *USR* values, for  ECG signal No. 101.

		M-OMP			SPGL1			M-FOCUSS
***USR***	Ψ	$\Psi_{\alpha 1}$	$\Psi_{\alpha 2}$	$\Psi_{\alpha 3}$	$\Psi_{\alpha 1}$	$\Psi_{\alpha 2}$	$\Psi_{\alpha 3}$	$\Psi_{\alpha 1}$
2	1.96	1.95	1.81	1.87	1.92	1.85	1.83	1.95
3	2.77	2.71	2.56	2.50	2.64	2.51	2.39	2.78
4	4.16	4.44	4.03	3.84	3.70	3.80	3.76	3.54
5	7.83	7.81	5.79	6.38	5.80	6.27	6.35	5.41
6	11.36	11.81	8.81	9.37	8.98	9.74	8.87	8.37
7	14.77	15.86	12.23	11.54	11.52	13.28	12.63	10.49
8	18.18	19.70	13.93	13.38	14.20	16.32	16.09	12.46
9	27.30	27.35	21.02	17.04	19.22	23.66	20.49	17.46
10	36.01	37.05	29.02	21.48	28.24	29.34	24.54	27.41

**Table 9 sensors-21-05282-t009:** Comparison of PRD values from the algorithms using the three dictionary matrices at several *USR* values, for  ECG signal No. 106.

		M-OMP			SPGL1			M-FOCUSS
***USR***	Ψ	$\Psi_{\alpha 1}$	$\Psi_{\alpha 2}$	$\Psi_{\alpha 3}$	$\Psi_{\alpha 1}$	$\Psi_{\alpha 2}$	$\Psi_{\alpha 3}$	$\Psi_{\alpha 1}$
2	1.66	1.63	1.50	1.55	1.65	1.59	1.56	1.65
3	2.34	2.35	1.87	2.01	2.32	2.11	2.12	2.34
4	3.86	3.50	2.52	3.04	3.39	3.00	2,95	3.21
5	6.09	4.95	3.15	3.91	5.11	4.17	4.60	4.36
6	8.33	6.63	3.97	5.36	7.04	6.01	6.96	6.04
7	11.00	8.75	5.06	7.01	9.23	8.98	9.74	7.76
8	14.06	10.36	6.23	8.96	11.42	10.88	12.11	9.44
9	18.46	13.64	9.09	12.70	15.62	15.17	17.20	12.68
10	22.40	17.53	14.78	16.87	20.10	21.54	21.34	17.13

**Table 10 sensors-21-05282-t010:** Comparison of PRD values from the algorithms using the three dictionary matrices at several *USR* values, for ECG signal No. 112.

		M-OMP			SPGL1			M-FOCUSS
***USR***	Ψ	$\Psi_{\alpha 1}$	$\Psi_{\alpha 2}$	$\Psi_{\alpha 3}$	$\Psi_{\alpha 1}$	$\Psi_{\alpha 2}$	$\Psi_{\alpha 3}$	$\Psi_{\alpha 1}$
2	2.49	2.50	2.30	2.47	2.31	2.18	2.13	1.83
3	3.13	3.24	2.80	3.13	2.90	2.76	2.65	2.47
4	5.06	4.55	3.68	4.42	4.31	3.88	3.80	3.49
5	8.02	5.67	4.76	5.68	5.95	5.14	5.42	5.24
6	11.83	7.06	5.88	6.90	7.73	6.61	7.21	7.85
7	14.73	8.89	7.19	8.31	9.56	8.45	9.26	10.27
8	16.81	9.60	7.72	9.60	10.80	9.84	11.30	12.12
9	21.17	10.90	9.02	12.33	13.22	12.72	13.62	14.74
10	26.74	13.64	10.06	14.43	17.11	16.15	16.07	18.23

**Table 11 sensors-21-05282-t011:** Comparison of PRD values from the algorithms using the three dictionary matrices at several *USR* values, for ECG signal no. 113.

		M-OMP			SPGL1			M-FOCUSS
***USR***	Ψ	$\Psi_{\alpha 1}$	$\Psi_{\alpha 2}$	$\Psi_{\alpha 3}$	$\Psi_{\alpha 1}$	$\Psi_{\alpha 2}$	$\Psi_{\alpha 3}$	$\Psi_{\alpha 1}$
2	1.15	1.10	1.00	1.09	1.05	0.99	0.99	1.15
3	1.75	1.55	1.27	1.48	1.41	1.35	1.35	1.17
4	2.97	2.15	1.72	2.39	1.96	1.93	2.03	1.6
5	6.30	3.07	2.48	4.32	3.22	3.19	3.62	2.37
6	11.15	4.13	3.13	6.57	4.82	6.02	6.79	3.70
7	15.96	5.35	5.59	8.67	9.l3	11.55	9.47	4.89
8	18.54	6.88	9.37	9.93	13.73	13.09	12.74	5.23
9	27.48	7.34	16.44	13.68	21.61	20.87	18.69	8.67
10	32.79	15.51	21.36	16.74	30.78	25.53	23.22	15.61

**Table 12 sensors-21-05282-t012:** Lowest and highest PRD values obtained by the M-OMP algorithm with $\Psi_{\alpha 2}$ among the considered signals vs. USR.

	*USR*								
Signal No.	2	3	4	5	6	7	8	9	10
105	1.39	1.69	2.06	2.50	3.21	3.54	4.11	5.58	7.84
101	1.81	2.56	4.03	5.79	8.81	12.23	13.93	21.02	29.02

**Table 13 sensors-21-05282-t013:** Performance comparison between the proposed method and the reconstruction techniques presented in [5,14] on signal number 117.

	*USR*	PRDN
Mamaganian et al. [5]	4	15
	10	>45
Fira et al. [14]	4	7.20
	8	10.96
	10	12.67
Proposed Method	4	3.83
	8	11.76
	10	17.73

## Data Availability

MIT-BIH Arrhythmia Database at https://www.physionet.org/content/mitdb/1.0.0/ (accessed on 23 November 2020).

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
