# Peer review of "A Dictionary Optimization Method for Reconstruction of ECG Signals after Compressed Sensing"

_sensors, 2021, doi:10.3390/s21165282_

Round 1

Reviewer 1 Report

This manuscript introduces a new approach for compressing ECG with a personalized dictionary. Three different optimization methods are evaluated to prepare the patient-specific dictionary. The compressive sensing is finally taken to reconstruct the ECG signal with a personalized dictionary.  The method is tested on the MIT-BIH Arrhythmia Database, and the results verify the M-OMP achieves overall the best performance. The whole paper is of interest and well prepared. But the authors need to address some major concerns as below. 

  1. In this paper, the PRD of the proposed algorithm with three different dictionary learning methods is evaluated under different USR values. All the results are lack comparison with existing work in this area. This is important to demonstrate the novelty of the proposed method over existing ones.
  2. The results are evaluated on several ECG signals with the No. 102. 103, 105, 107, 122. Why don't the authors make an assessment of the overall data set, so as to give a statistically significant evaluation

Author Response

The authors thank the reviewer for the useful comments which allowed to improve the paper. Point-to-point response to the reviewers comments is reported in the attached file.

Reviewer 2 Report

  1. The entire introduction section needs to be revised to present the novelty of the proposed method and clearly state the problems it specifically addresses. In particular, the text between lines 43 and 97 provide a mere background and does not highlight the relevance of the proposed method/solution to the problem statement.
  2. Please cite any references that take a similar optimization approach for different signal and how your work is different from that. Only applying on a different signal (ECG in this case) is not sufficient to make the work novel.
  3. The cases [re-]described between lines 366 and 412 needs to be moved to the previous section where they are initially introduced. This way the results section will only demonstrate the related results and will not walk the reader again through the methods.
  4. A pictorial presentation of the overall process and the methods can help with the understanding of the process.
  5. Please provide information about time between ECG segments picked for training phase and reconstruction phase for each ECG set.
  6. In lines 167 to 170 what “sensor nodes” refer to and how that is related to pre-recorded ECGs used in the study? If it is elaborated in Reference 20 and the understanding of it completely relies on that reference, I suggest a brief description that save a reader to go to reference 20 might be beneficial.
  7. In Figure 2 why there are two “frame splitting” blocks? Is it a typo?
  8. In Figure 2 how the block “CS based compression” decides which output arrow it should serve? Are the outputs conditional or simultaneous? If simultaneous, use a single output from the block and branch it to the other blocks (“Nyquist Rate ECG Signal Acquisition” and “Wireless Communication” cloud). Otherwise, please indicate when which output is generated by use of conditional blocks.
  9. Number of ECG signal sets of only 5 is too small for conclusive results.
  10. The figures need their legend to describe the blue line and red stars. This should be in addition to the in-text description to help readability of the figures without the need to go back to the text to find the explanations.
  11. Please demonstrate the range of performance across ECG signals. Possibly by including results from at least two ECG sets one resulting to best performance and one resulting to the worst performance.

Author Response

(The authors gave the same response as above.)

Round 2

Reviewer 1 Report

The paper quality was clearly improved. 

There still exist some grammar issues and the quality of some descriptions need to be improved in the revised manuscript.  For example, 
1) In Figure 2, the mathematical formula is not well edited.
2)   In line 278-279, 'So the tasks to perform in order to achieve the lowest PRD  are the following'

Author Response

The authors thank once again the reviewer for the comments which allowed to further improve the paper. 

Attached you will find the point-to-point response to the reviewers' comments.

Best regards

The authors
